

# Deep ocular tumor classification model using cuckoo search algorithm and Caputo fractional gradient descent

Abduljlil Abduljlil Ali Abduljlil Habeb[1], Ningbo Zhu[1,2], Mundher Mohammed Taresh[1] and Talal Ahmed Ali Ali[1]

[1] College of Computer Science and Electronic Engineering, Hunan University, Changsha, Hunan, China
[2] Research Institute, Hunan University, Chongqing, Chongqing, China

Corresponding author
Ningbo Zhu, quietwave@hnu.edu.cn

## ABSTRACT

While digital ocular fundus images are commonly used for diagnosing ocular tumors, interpreting these images poses challenges due to their complexity and the subtle features specific to tumors. Automated detection of ocular tumors is crucial for timely diagnosis and effective treatment. This study investigates a robust deep learning system designed for classifying ocular tumors. The article introduces a novel optimizer that integrates the Caputo fractional gradient descent (CFGD) method with the cuckoo search algorithm (CSA) to enhance accuracy and convergence speed, seeking optimal solutions. The proposed optimizer's performance is assessed by training well-known Vgg16, AlexNet, and GoogLeNet models on 400 fundus images, equally divided between benign and malignant classes. Results demonstrate the significant potential of the proposed optimizer in improving classification accuracy and convergence speed. In particular, the mean accuracy attained by the proposed optimizer is 86.43%, 87.42%, and 87.62% for the Vgg16, AlexNet, and GoogLeNet models, respectively. The performance of our optimizer is compared with existing approaches, namely stochastic gradient descent with momentum (SGDM), adaptive momentum estimation (ADAM), the original cuckoo search algorithm (CSA), Caputo fractional gradient descent (CFGD), beetle antenna search with ADAM (BASADAM), and CSA with ADAM (CSA-ADAM). Evaluation criteria encompass accuracy, robustness, consistency, and convergence speed. Comparative results highlight significant enhancements across all metrics, showcasing the potential of deep learning techniques with the proposed optimizer for accurately identifying ocular tumors. This research contributes significantly to the development of computer-aided diagnosis systems for ocular tumors, emphasizing the benefits of the proposed optimizer in medical image classification domains.

## INTRODUCTION

Ocular tumors refer to irregular growths that can develop in various parts of the eye. These growths are categorized as either benign or malignant, reflecting the degree of potential harm they can cause. Even though benign tumors may seem less harmful, timely attention

is crucial, as they can still affect vision. Conversely, malignant tumors represent a more serious threat since they have the potential to metastasize to other organs (*Grishina, Kim & Izotova, 2023*), resulting in myriad health issues. Therefore, the detection and treatment of ocular tumors are crucial in order to preserve vision and for the purpose of preventing metastasis, thereby ensuring the avoidance of potential side effects.

Fundus imaging is a prevalent non-invasive technique in ophthalmology that is predicated on symptom recognition for diagnosis. However, its effectiveness can be limited by variations and individual awareness among ophthalmologists (*Honavar, 2021*; *Pogosova, 2022*). Specialized training and expertise are necessary for the accurate interpretation of fundus images, a requirement that may be deficient in resource-limited settings (*Ramírez-Ortiz et al., 2017*). Moreover, the diverse clinical features of these tumors pose a formidable challenge (*Neupane, Gaudana & Boddu, 2018*; *Liu et al., 2019*), especially in the course of identifying small or deep-seated tumors (*Gündüz & Tetik, 2023*; *Moothedath, Seth & Chawla, 2023*; *Manjandavida et al., 2019*). As a matter of fact, locating these tumors can prove challenging even for experienced medical professionals.

In recent times, machine and deep learning techniques have gained prominence in medical diagnosis, involving the training of classification models on specific datasets to identify a range of ophthalmic conditions, including various ocular diseases (*Nawaz et al., 2022*; *Akter et al., 2022*; *Atwany, Sahyoun & Yaqub, 2022*; *Das, Biswas & Bandyopadhyay, 2022*; *Kadry et al., 2022*; *Jin et al., 2022*) and tumors (*Goswami, 2021*; *Kaliki et al., 2023*; *Kumar et al., 2023*).

Nevertheless, the preferred technique in such tasks is transfer learning due to the constrained size of medical datasets. This approach involves leveraging pre-trained models, such as Vgg16 (*Simonyan & Zisserman, 2014*), GoogLeNet (*Szegedy et al., 2015*), and AlexNet (*Russakovsky et al., 2015*). The architecture of GoogLeNet introduced inception modules, which facilitated efficient usage of computing resources (*Bilal et al., 2022*). Vgg16's deep architecture is beneficial for diverse image classification tasks, including medical image classification (*Albashish et al., 2021*). In fundus image analysis, AlexNet excels in learning robust features, contributing to enhanced overall performance (*Deepika & Shivakumar, 2021*). These models undergo training on extensive datasets for the purpose of diagnosing the targeted disease. This training process incorporates optimization techniques like stochastic gradient descent with momentum (SGDM) and adaptive momentum estimation (ADAM) to fine-tune the weights of specific layers in the pre-existing model. This adaptation facilitates the transfer of knowledge acquired from prior tasks to effectively address the new diagnostic task. Additionally, certain authors endeavor to enhance the convergence rate through optimization methodologies grounded in fractional calculus (*Pu et al., 2013*; *Sheng et al., 2020*; *Taresh et al., 2022*), which employ fractional derivatives as a generalization for conventional derivatives. However, the diverse clinical features of ocular tumors and limited data availability pose significant challenges, leading to issues such as overfitting and hyperparameter sensitivity when using these optimization algorithms (*Sengupta et al., 2018*). These impediments primarily stem from the gradient-based nature of the optimization techniques, which can converge to local minima in highly non-convex problems.

In response to the limitations of gradient-based optimizers, researchers have investigated the optimization of their classification models through global search techniques by using meta-heuristics algorithms (*Ólafsson, 2006*; *Gogna & Tayal, 2013*; *Nesmachnow, 2014*). The objective of these techniques is to discover global optimal solutions without depending on gradient information. Meta-heuristic algorithms frequently work with populations, commencing with a set of random solutions and progressively converging toward the global optimum through diverse phenomena. Typically, their search process comprises two phases: exploration, during which they randomly explore new solutions across the entire search space, and exploitation, where the emphasis is on searching around the best solution acquired thus far. Achieving an optimal balance between exploration and exploitation is essential to avoid low accuracy and the risk of becoming trapped in local optima.

While meta-heuristics excel in various engineering and scientific problems, due to the high nonlinearity and the vast size of the search space with numerous parameters, their performance tends to be suboptimal in optimizing deep models. To surmount these challenges, recent efforts have placed the emphasis on integrating ADAM into meta-heuristic techniques such as cuckoo search (CSA) (*Mohsin, Li & Abdalla, 2020*) and beetle antenna search (BAS) (*Khan et al., 2020*), to guide the search towards feasible regions. While the combination of gradient-based methods with meta-heuristics has demonstrated promising results, additional research is imperative to optimize their application across diverse scenarios.

The objective of this article is to improve the performance of the (CSA) in optimizing deep models by integrating a fractional gradient descent mechanism into the heuristic search engine. The noteworthy aspect of fractional gradient descent lies in its capacity to maintain a memory of past iterations and adapt its search direction accordingly. This characteristic has the potential to result in a more efficient and resilient optimization process. To achieve that goal, we employ the Caputo fractional gradient descent method (CFGD) (*Shin, Darbon & Karniadakis, 2021*), which renowned for its proven monotonicity and convergence properties (*Wang et al., 2017*; *Chen & Zhao, 2019*; *Taresh et al., 2022*; *Shin, Darbon & Karniadakis, 2023*).

Furthermore, our approach leverages the unique property of Caputo's definition, wherein the fractional differential of a constant function is consistently 0. This alignment with the principles of integer-order calculus enhances the applicability of the Caputo definition in engineering problem-solving. The incorporation of the Caputo formulation enhances the memory-efficient capabilities of fractional gradient descent, particularly when calculating fractional-order derivatives in our task. As a result, the overall effectiveness of our optimization technique is heightened. Furthermore, the main reason behind choosing CSA is its promising performance in medical image classification tasks (*Goel, Gaur & Jain, 2015*; *Guerrout et al., 2020*; *Mohsin, Li & Abdalla, 2020*). Notably, the proposed optimization technique is applied to well-established pre-trained models known for their efficacy in fundus image classification (*Shaik & Cherukuri, 2022*; *Velpula & Sharma, 2023*; *Salma, Bustamam & Sarwinda, 2021*), specifically Vgg16, AlexNet, and GoogLeNet. To the best of our knowledge, this is the first work introducing the application of optimization

technique to the ocular tumor classification using deep learning models. The significant contributions made in this article are manifolded.

1. We propose a new optimizer based on integrating a Caputo fractional gradient descent (CFGD) to CSA; hereafter referred to as CSA-CFGD. This integration enables a more sophisticated exploration of the search space and an improved exploitation around the most feasible region by considering the historical information of the objective function.
2. We propose a novel approach for ocular tumors classification by utilizing data augmentation and training pre-trained models, namely Vgg16, AlexNet, and GoogLeNet, on a dataset of fundus images containing ocular tumors. The data augmentation techniques include random rotation, horizontal and vertical shifts, shear transformation, zoom, flips, and brightness adjustment. Through the application of data augmentation, we substantially augment the diversity of the training data, leading to enhanced model generalization and improved performance in real-world scenarios.
3. We carried out twenty independent runs of training of each pre-trained model, comprehensively comparing the obtained models, offering valuable insights into their performance in ocular tumor classification and their suitability for the task.
4. Using the pre-trained models employed in this study, we compared our proposed optimization algorithm with existing algorithms in the literature. We conducted a comprehensive investigation, including average case performance studies and statistical hypothesis tests such as parametric (two sample $t$-test) and non-parametric (Wilcoxon rank-sum test) analyses.

The rest of this article is structured as follows: "The New Optimizer" provides a comprehensive outline of the optimizer that is being proposed. Next, the section titled "Ocular Tumor Classification Neural Network" elucidates the neural network that was developed specifically for the classification of ocular tumors. The setup for the simulation experiments is described in the "Simulation Experiments Setup" section. The "Results and Discussion" section present and explicate the results as well as discussion of the experiments. The section titled "Conclusion" provides a summary of our conclusions and suggests possible directions for further investigation.

## THE NEW OPTIMIZER

For a real-valued function $f(\mathbf{x})$, $\mathbf{x} = [x_0, x_1, \cdots, x_d] \in \mathbb{R}^d$, it is often required to find the value of $\mathbf{x}$ that minimize $f(\mathbf{x})$, *i.e.*,

$$\min_{\mathbf{x} \in \mathbb{R}^d} f(\mathbf{x}). \tag{1}$$

The conventional gradient decent method has the following iterative rule for $\mathbf{x}$,

$$\mathbf{x}^{(k+1)} = \mathbf{x}^{(k)} - \eta_k \cdot \nabla_{\boldsymbol{x}} f\left(\mathbf{x}^{(k)}\right), \tag{2}$$

where $k$ denotes the number of iteration and $\eta_k$ is the learning rate at the $k$th iteration. Such a method searches for an optimal solution by taking discrete steps in the direction of steepest descent. This method often converges linearly to a stationary point provided that the learning rates are appropriately chosen.

## Cuckoo search algorithm

The cuckoo search algorithm (CSA) is a meta-heuristic algorithm inspired by the brood parasitism behavior of cuckoo species. It incorporates lévy flights in order to improve its performance. In CSA, each cuckoo lays just one egg in a randomly selected nest, and nests with high-quality eggs are retained for future generations. The host bird can choose to either discard the egg or build a new nest, and it discovers cuckoo eggs with a probability $p_a \in [0, 1]$. Unlike other meta-heuristic algorithms, the CSA method generates two populations of potential solutions using lévy flight and random walk. The population resulting from lévy flight exhibits exploration in the search space, with individuals being notably diverse due to the lévy function's stochastic nature. CSA employs the switching parameter $p_a$ to combine global and local random walks. The global random walk uses lévy flight operation $Levy(\lambda) \sim u = k^{-\lambda}$, $1 < \lambda \leq 3$, to explore the search space, *i.e.,*

$$\mathbf{x}_j^{(k+1)} = \mathbf{x}_j^{(k)} + \beta \otimes Levy(\lambda). \tag{3}$$

In the local random walk, two solutions, $\mathbf{x}_p^k$ and $\mathbf{x}_q^k$, are randomly selected through permutation. The new position of the $j$th nest at the $k$th iteration is calculated using Eq. (4), which involves the current position $\mathbf{x}_i^{(k)}$, step size $\beta$, step scaling factor $s$, Heaviside function $H(u)$, probability of discovering a cuckoo egg $p_a$, element-wise product of vectors $\otimes$, and a random number drawn from a uniform distribution $v$.

$$\mathbf{x}_j^{(k+1)} = \mathbf{x}_j^{(k)} + \beta s \otimes H(p_a - v) \otimes (\mathbf{x}_p^{(k)} - \mathbf{x}_q^{(k)}). \tag{4}$$

The algorithm's basic steps, including cuckoo selection, solution generation, evaluation, nest replacement, abandonment, as well as update, are summarized in Algorithm 1. The stochastic Eq. (4) represents a random walk, where the next location/state depends on the current location and the transition probability. However, in CS, a significant portion of new solutions should be generated through far-field randomization to ensure sufficient exploration of the search space and to avoid getting trapped in local optima.

---

**Algorithm 1** Cuckoo Search via Lvy Flights

---

**begin**
Objective function $f(\mathbf{x}), \mathbf{x} = (x_1, ..., x_d)^T$
Generate initial population of $n$ host nests $\mathbf{x}_j$ ($j = 0, 1, \cdots, n$)
**while** ($k <$ MaxGeneration) or (stop criterion) **do**
    Get a cuckoo randomly by Lvy flights using (3)
    Evaluate its solution quality or objective value $f(\mathbf{x}_j)$
    Choose a nest among $n$ (say, $l$) randomly
    **if** ($f(\mathbf{x}_j) < f(\mathbf{x}_l)$) **then**
        Replace $l$ by the new solution $j$
    **end if**
    A fraction ($p_a$) of worse nests are abandoned
    New nests/solutions are built using (4)
    Keep best solutions (or nests with quality solutions)
    Rank the solutions and find the current best
    Update $k \leftarrow k + 1$
**end while**
Postprocess results and visualization
**end**

---

In CSA, the step size plays a pivotal role and necessitates vigilant monitoring to discern the search area relevant to the practical problem. In our approach, we predefine these step

size values, and they remain constant across generations. However, this fixed nature can pose challenges, potentially causing the algorithm to become entrenched in local optima, thereby complicating the task of discovering optimal solutions.

CSA utilizes the lévy flight strategy, enabling cuckoo nests to traverse *via* a combination of short and sporadic long-distance cooperative random searches. It is due to this particular mode that CSA exhibits a significant and unpredictable leap during the search process. Consequently, the search in the surrounding area of each cuckoo nest is not sufficiently robust, leading to a slow convergence speed and inadequate convergence accuracy for CSA. The gradient direction, which represents the direction of maximum value change, is indicated by a vector that comes from the $\mathbf{x}_{worst}$ point and terminates at the $\mathbf{x}_{best}$ point.

For $i = 1, \ldots, d$, we define the functions $f_{i,x} : \mathbb{R} \to \mathbb{R}$ by $f_{i,x}(y) = f(x + (y - x_i)e_i)$, where $e_i$ represents the vector in $\mathbb{R}^d$ with a 1 in the $i$-th coordinate and 0's elsewhere. For a vector $c_k = [c_0, c_1, c_2, \ldots, c_d] \in \mathbb{R}^d$, we define the Caputo fractional gradient of $f$ by

$$c^{kC}\nabla_x^\alpha f(x^{(k)}) = \left(c_1^{\text{Caputo}}D_x^\alpha f_{1,x}(x_1), \ldots, c_d^{\text{Caputo}}D_x^\alpha f_{d,x}(x_d)\right) \in \mathbb{R}^d. \tag{5}$$

We can now introduce a Caputo fractional gradient descent method (CFGD) in the following manner: Starting at an initial point $x^{(0)}$, the $k$-th iterated solution is updated by

$$\mathbf{x}^{(k+1)} = \mathbf{x}^{(k)} - \eta_k \cdot \mathbf{c}^{kC}\nabla_{\mathbf{x}}^\alpha f\left(\mathbf{x}^{(k)}\right), \quad k = 0, 1, \ldots, \tag{6}$$

where $\alpha \in (0, 1)$, $x^{(k)} = (x_i^k)$, $c^k = (x_i^k)$, $\gamma_k \in \mathbb{R}$ and $_{c^k}^{C}\nabla_x^\alpha f(x^{(k)})$ is expressed as:

$$\text{diag}\left(c_i^{k\,\text{Caputo}}D_x^\alpha I(x_i^k)\right)^{-1}\left[c^{kC}\nabla_x^\alpha f(x^{(k)}) + \gamma_k\,\text{diag}(|x_i^k - c_i^k|)c^{kC}\nabla_x^{\alpha+1}f(x^{(k)})\right].$$

Let $f(x)$ be a real-valued sufficiently smooth function defined on $\mathbb{R}^d$, where $c_\alpha = (1 - \alpha)2^{-(1-\alpha)}$, we have

$$\left(c^{kC}\nabla_x^\alpha f(x^{(k)})\right)_i \tag{7}$$

$$= c_{\alpha_k}\int_{-1}^1 f_{i,x}'(\Delta_i^k(1+u) + c_i^k)(1-u)^{-\alpha_k}\,du + c_{\alpha_k}\gamma_k|x_i^k - c_i^k|\int_{-1}^1 f_{i,x}''(\Delta_i^k(1+u)+c_i^k)(1-u)^{-\alpha_k}\,du,$$

We observe that Eq. (7) involves integrals that can be accurately evaluated by the Gauss-Jacobi quadrature. Let $\{(\lambda_m, u_m)\}_{m=1}^v$ be the Gauss-Jacobi quadrature rule of $v$ points. Then, $\mathbf{c}^{kC}\nabla_{\mathbf{x}}^\alpha f\left(\mathbf{x}^{(k)}\right)$ can be approximated using the Gauss-Jacobi quadrature formula as:

$$\left(\mathbf{c}^{kC}\nabla_{\mathbf{x}}^\alpha f\left(\mathbf{x}^{(k)}\right)\right)_i = C_{\alpha_k}\sum_{m=0}^M \lambda_m f_i'(\Delta_i^k(1+u_m) + c_i^k) \quad +$$

$$C_{\alpha_k}\gamma_k\left|x_i^k - c_i^k\right|\sum_{m=0}^M \lambda_m f_i''(\Delta_i^k(1+u_m) + c_i^k). \tag{8}$$

The steepest descent direction of a locally smoothed original objective function is the generic CFGD. Incorporating information about the local gradient and curvature of the objective function, this formula provides a local approximation of the Caputo fractional gradient. The parameters $\alpha$, $\gamma$, and the weights and nodes of the quadrature formula $(\lambda_m, u_m)$ collectively control the method's behavior, thus achieving a balance between

exploration and exploitation in the optimization process. To enhance exploration in feasible regions, we incorporate the Caputo fractional gradient given in Eq. (8) into Eq. (3), which yields

$$\mathbf{x}_j^{(k+1)} = \mathbf{x}_j^{(k)} + \beta_k \otimes Levy(\lambda)_{\mathbf{c}^k}^C \nabla \mathbf{x}^\alpha f\left(\mathbf{x}_{best}^{(k)}\right), \tag{9}$$

where $\mathbf{c}^k$ is computed as the mean of the three solutions $\mathbf{x}_1$, $\mathbf{x}_2$ and $\mathbf{x}_3$, *i.e.*,

$$\mathbf{c}^k = \frac{\mathbf{x}_1 + \mathbf{x}_2 + \mathbf{x}_3}{3}, \tag{10}$$

where $\mathbf{x}_{1,2,3} = \mathbf{x}_j - r_{1,2,3}(\mathbf{x}_{best} - \mathbf{x}_j^k)$, and $r_{1,2,3}$ denote random numbers that uniformly distributed over $[0, 2]$. Such random numbers augment the exploration of the new regions and help avoid local minima stagnation. Similarly, we incorporate the Caputo fractional gradient descent given in Eq. (8) into Eq. (4) to improve the exploitation around the best solution obtained so far, which yields

$$\mathbf{x}_j^{(k+1)} = \mathbf{x}_j^{(k)} + \beta s \otimes H(p_a - v) \otimes_{\mathbf{c}^k}^C \nabla_\mathbf{x}^\alpha f\left(\mathbf{x}_{best}^{(k)}\right), \tag{11}$$

where $\mathbf{c}^k$ is assigned to $\mathbf{x}_{worst}$.

The integration of CFGD into the cuckoo search algorithm (CSA) necessitates the incorporation of CFGD components, particularly the Caputo fractional gradient, at strategic points within the CSA algorithm. This integration aims to boost the overall performance of the optimization process by leveraging the distinctive advantages offered by both CSA and CFGD. Through meticulous design, the integration is tailored to capitalize on these specific strengths. The primary objective is to achieve higher convergence and more efficient exploration of the solution space by combining CSA's global search capabilities with CFGD's gradient-based method. The introduction of additional components, such as random numbers and CFGD-specific rules, introduces new computational tasks that could potentially impact the overall efficiency of the algorithm. While this integration is designed to be modular and controllable, it is crucial to recognize that this diversity may lead to additional computational expenses. The pseudocode for CSA-CFGD is provided below in Algorithm 2.

## THE PROPOSED OCULAR TUMORS CLASSIFICATION NEURAL NETWORK

The proposed ocular tumor classification approach leverages pretrained convolutional neural network (CNN) models, including Vgg16, AlexNet, and GoogLeNet, for feature extraction. The feature extraction process is denoted as follows:

$$\mathbf{H}^{(l)} = \sigma\left(\sum_{i=1}^{N^{(l-1)}} \mathbf{W}_i^{(l)} * \mathbf{H}_i^{(l-1)} + \mathbf{b}^{(l)}\right), \tag{12}$$

where $\mathbf{H}^{(l)}$ represents the feature map at layer $l$, $\sigma$ is the rectified linear unit (ReLU) activation function, $\mathbf{N}^{(l-1)}$ signifies the number of neurons or nodes in the input feature map $\mathbf{H}^{(l-1)}$ at layer $(l-1)$ of the neural network, $\mathbf{W}_i^{(l)}$ represents weights, $\mathbf{H}_i^{(l-1)}$ is the

---

**Algorithm 2** CSA-CFGD

```
begin
    Objective function f(x), x = (x₁,...,x_d)ᵀ
    Generate initial population of n host nests x_j (j = 0, 1, ···, n)
    while (k <MaxGeneration) or (stop criterion) do
        Get a cuckoo randomly by Lvy flights using (9)
        Evaluate its solution quality or objective value f(x_j)
        Choose a nest among n (say, l) randomly
        if (f(x_j) < f(x_l)) then
            Replace l by the new solution j
        end if
        A fraction (p_a) of worse nests are abandoned
        New nests/solutions are built using (11)
        Keep the best solutions (or nests with quality solutions)
        Rank the solutions and find the current best
        Update k ← k + 1
    end while
    Postprocess results and visualization
end
```

---

input feature map, and $\mathbf{b}^{(l)}$ is the bias. The final feature map was flattened to input a fully connected neural network classifier during training. We replaced the top layers in each model with two fully connected layers, denoted as $\mathbf{H}_{fc1}$ and $\mathbf{H}_{fc2}$. These layers use ReLU activation, which effectively improves the convergence and efficiency of neural networks, especially in image classification. They are expressed as follows:

$$\mathbf{H}_{fc1} = \sigma\left(\mathbf{W}_{fc1} \cdot \mathbf{H}^{(L-1)} + \mathbf{b}_{fc1}\right), \tag{13}$$

$$\mathbf{H}_{fc2} = \sigma\left(\mathbf{W}_{fc2} \cdot \mathbf{H}_{fc1} + \mathbf{b}_{fc2}\right), \tag{14}$$

where $L$ denotes the total number of layers, $\mathbf{W}_{fc1}$, $\mathbf{W}_{fc2}$ repesent the weights of the fully connected layers, and $\mathbf{b}_{fc1}$,$\mathbf{b}_{fc2}$ signify the biases. Finally, the output layer employs a sigmoid function for binary classification, denoted as $\mathbf{Y}$, representing the predicted output probabilities:

$$\mathbf{Y} = \sigma\left(\mathbf{W}_{fc} \cdot \mathbf{H}_{fc2} + \mathbf{b}_{fc}\right). \tag{15}$$

Using ReLU in the hidden layers allows the network to efficiently learn complex features. In addition, the use of sigmoid in the output layer provides a probability-like output suitable for binary classification. In tasks where the output is binary (0 or 1), the binary cross-entropy loss $L$, also known as log loss, is a standard choice for the loss function. This loss function imposes penalties for misclassifications by quantifying the disparity between predicted probabilities and actual labels, resulting in effective training for binary classification scenarios. The formula for binary cross-entropy loss is:

$$L(y, \hat{y}) = -\frac{1}{N} \sum_{i=1}^{N} \left[ y_i \cdot \log(\hat{y}_i) + (1 - y_i) \cdot \log(1 - \hat{y}_i) \right], \tag{16}$$

where $N$ denotes the number of samples $y_i$ is the true label for the $i$-th sample and $\hat{y}_i$ is the predicted probability for the $i$-th sample.

To optimize the neural network, we employed the CSA-CFGD as our optimization algorithm, helping us obtain optimal weights and biases for the network. The optimization

---

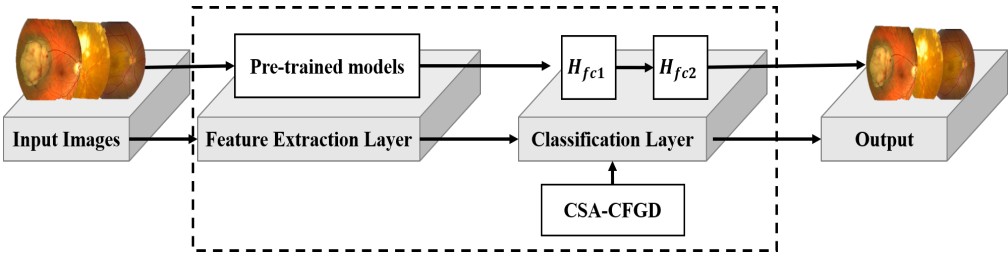

**Figure 1** **Outline of the proposed model.** Image source credits: This image was originally published in the Retina Image Bank® website. Alex P. Hunyor, MD. choroidal melanoma Retina Image Bank. 2013, 3101. © the American Society of Retina Specialists https://imagebank.asrs.org/file/3101/choroidal-melanoma-case-4-partly-amelanotic. This image was originally published in the Retina Image Bank® website. Jason S. Calhoun. choroidal hemangioma Retina Image Bank. 2013, 8214. © the American Society of Retina Specialists https://imagebank.asrs.org/file/8214/cavernous-choroidal-hemangioma.

process can be represented as follows:

Update $\mathbf{W}_{fc1}$ and $\mathbf{b}_{fc1}$ using CSA-CFGD, (17)

Update $\mathbf{W}_{fc2}$ and $\mathbf{b}_{fc2}$ using CSA-CFGD. (18)

Figure 1 illustrates the architecture of our proposed CAS-CFGD based CNN for the ocular tumor classification from fundus images. Compared to the existing optimization algorithms in the literature, we expect to achieve superior results in ocular tumor identification using the proposed optimizer.

## SIMULATION EXPERIMENTS SETUP

In this section, we enunciate the setup as well as implementation details of the simulation experiments conducted to evaluate the performance and effectiveness of ocular tumors classification model using the CSA-CFGD. The experiments were performed using MATLAB 9.0 on an Intel(R) Core(TM) i7-4510U CPU @ 2.6 GHz with 8 GB RAM.

To address the issue of overfitting, a 10-fold cross-validation approach is employed, utilizing 70% of the entire dataset for training. The experimental data comprises two classes, namely benign and malignant, with each class containing 200 patches, resulting in a total of 400 patches. It is collected from the Retina Image Bank (*American Society of Retina Specialists , 2022*) created by the American Society of Retina Specialists in It allows ophthalmologists and photographers from all parts of the globe to come together and share real-life patient cases online. Figure 2 displays image samples of benign and malignant of the ocular using fundus image. Ophthalmologists with specialized expertise in the field meticulously reviewed and verified the images to ensure their quality and accuracy. These images were then utilized to deepen understanding and knowledge of retinal diseases.

Numerous augmentation techniques are applied to the training data, encompassing random rotation within −30 to 30 degrees, random horizontal and vertical shifts within

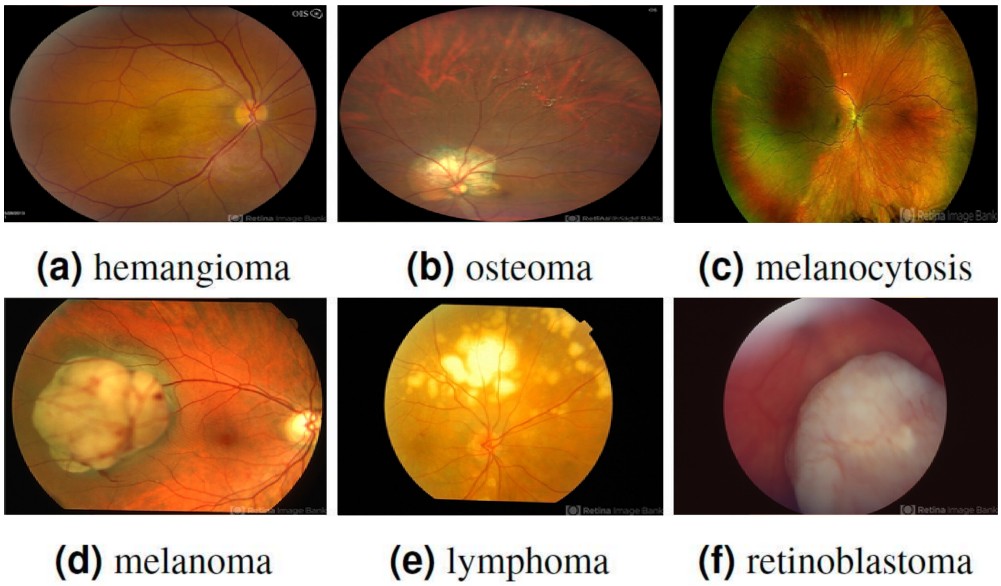

**Figure 2 Displaying three benign ocular tumors in the first row and three malignant ocular tumors in the second row.** Image source credits: (A) This image was originally published in the Retina Image Bank® website. Jason S. Calhoun. choroidal hemangioma Retina Image Bank. 2013, 8214. © the American Society of Retina Specialists https://imagebank.asrs.org/file/8214/cavernous-choroidal-hemangioma. (B) This image was originally published in the Retina Image Bank® website. Gregg T. Kokame, MD, MMM, FASRS. choroidal-osteoma Retina Image Bank. 2020, 47685. © the American Society of Retina Specialists https://imagebank.asrs.org/file/47685/choroidal-osteoma. (C) This image was originally published in the Retina Image Bank® website. John S. King, MD. choroidal melanocytosis Retina Image Bank. 2019, 39310. © the American Society of Retina Specialists https://imagebank.asrs.org/file/39310/isolated-choroidal-melanocytosis-montage. (D) This image was originally published in the Retina Image Bank® website. Alex P. Hunyor, MD. choroidal melanoma Retina Image Bank. 2013, 3101. © the American Society of Retina Specialists https://imagebank.asrs.org/file/3101/choroidal-melanoma-case-4-partly-amelanotic. (E) This image was originally published in the Retina Image Bank® website. Mallika Goyal, MD. Choroidal Lymphoma Retina Image Bank. 2012, 2154. © the American Society of Retina Specialists https://imagebank.asrs.org/file/2154/choroidal-lymphoma. (F) This image was originally published in the Retina Image Bank® website. Gary R. Cook, MD, FACS. Retinoblastoma Retina Image Bank. 2019, 29815. © the American Society of Retina Specialists https://imagebank.asrs.org/file/29815/bilateral-retinoblastoma.

−0.1 to 0.1, random shear transformation within −0.2 to 0.2, random zoom within 0.8 to 1.2, random horizontal and vertical flips, and random brightness adjustment within 0.8 to 1.2. The implementation of these augmentation techniques improves the diversity of the data, which, in turn, enhances the generalization and performance of the model in real-world situations. The carefully visualized augmented images in Fig. 3 demonstrate the diverse transformations applied to the original images, thereby exemplifying the efficacy of the data augmentation process in our research.

Meta-heuristic techniques rely on specific parameters that have a significant impact on their performance. The choice of hyper-parameters significantly influences the model's performance. Table 1 provides an overview of the salient parameters employed in these configurations. A population size of 50 is maintained throughout the experiments for all

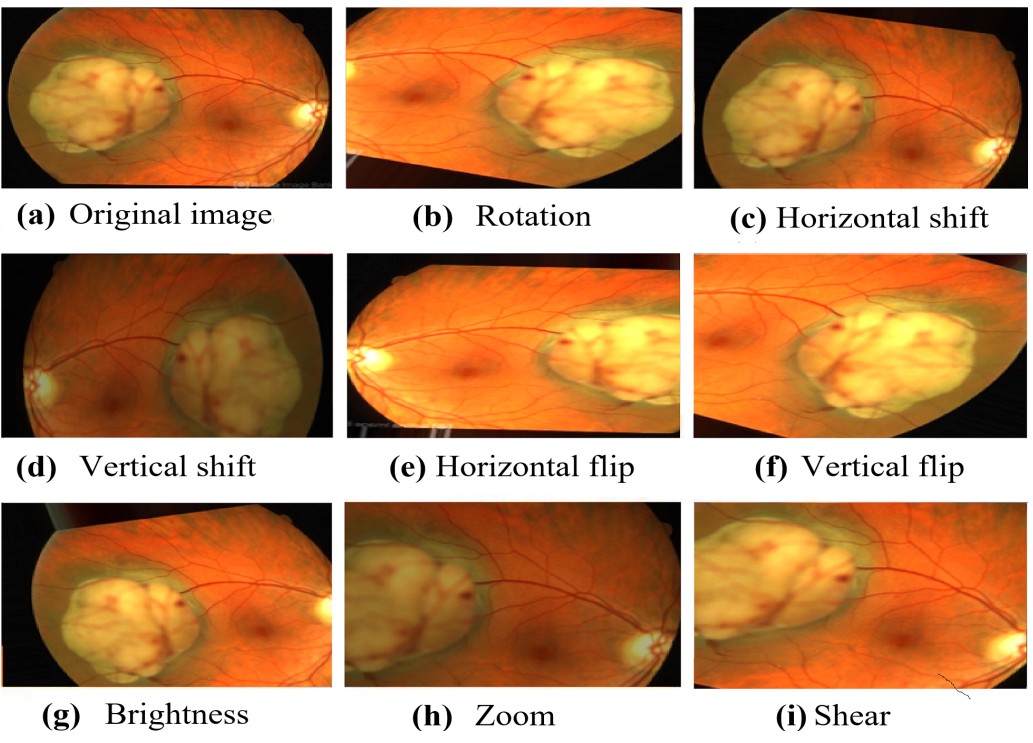

**(a)** Original image     **(b)** Rotation     **(c)** Horizontal shift

**(d)** Vertical shift     **(e)** Horizontal flip     **(f)** Vertical flip

**(g)** Brightness     **(h)** Zoom     **(i)** Shear

**Figure 3** **The resulted augmented image.** Image source credit: This image was originally published in the Retina Image Bank® website. Alex P. Hunyor, MD. choroidal melanoma Retina Image Bank. 2013, 3101. ©️ the American Society of Retina Specialists https://imagebank.asrs.org/file/3101/choroidal-melanoma-case-4-partly-amelanotic.

**Table 1** **Hyper-parameters of CNN models used in this study.**

| Hyper-parameter | Value |
|---|---|
| Learning rate (LR) | $10^{-4}$ |
| Epochs | 6 (400 iterations in each epoch) |
| Batch size (BS) | 32 |
| Number of nodes | 512 |
| Activation function | ReLU for the $\mathbf{H}_{fc}$ layers and sigmoid for the output layer |
| Nodes | 1024 for $\mathbf{H}_{fc1}$ and 512 for $\mathbf{H}_{fc2}$ |
| Population size | 50 |
| Generations | 100 |
| The probability | 0.25 (for CSA algoirthms) |
| Antennae length | 10 (for BAS algorithm) |

algorithms. A total of 100 generations are subsequently run. The CSA utilize a probability of 0.25 for cuckoo egg laying. In the case of the beetle antennae search, the antennae length is set to 10, inspired by *Khan et al. (2020)*.

Moreover, neural network configurations play a pivotal role in ensuring the success of deep learning experiments. The architecture of the neural network includes fully connected

layers, where the number of nodes plays a critical role in capturing complex patterns. To ensure effective learning, we set the learning rate (LR) to $10^{-4}$, controlling the step size during the optimization process. The training process was conducted over six epochs, each encompassing 400 iterations. This framework allows the algorithm to progressively refine its population, exploring and exploiting the solution space to uncover more optimal solutions. A batch size (BS) of 32 was employed, specifying the number of samples processed in each iteration. Furthermore, the specific number of nodes in each hidden layer may vary depending on the implementation. In our study, we chose 1,024 nodes for the first hidden layer and 512 for the second, aligning with the size and complexity of the datasets used in our experiments. This node configuration follows established practices in deep learning, proving effective in capturing intricate patterns within the data. The input images are resized to 224 × 224 for Vgg16 and GoogLeNet and 256 × 256 for AlexNet.

The performance of the proposed CSA-CFGD-based CNN isthen analyzed and compared with existing methods to classify ocular tumors. Besides the original CSA, we chose to compare our optimizer with well-known optimizers, including SGDM, and ADAM, as well as the optimizers proposed in *Sheng et al. (2020)* for CFGD, *Khan et al. (2020)* for BAS-ADAM, and *Mohsin, Li & Abdalla (2020)* for CSA-ADAM. In order to conduct this comparison, we leveraged a dataset to train the selected pre-trained model using these optimizers and subsequently compared the obtained results.

## RESULT

This section provides a comprehensive analysis of the proposed optimizer and existing optimization algorithms. In this study, the CSA algorithm is enhanced by integrating CFGD. The error function gradient is calculated experimentally using various fractional $\alpha$-order derivatives, where $0 < \alpha \leq 1$. Significant efforts have been invested in determining the optimal value for the order at which the algorithm demonstrates rapid convergence. It is noteworthy that when $\alpha = 1$, it denotes the use of traditional optimizers without integrating CFGD, thus serving as the standard CSA configuration for our proposed optimizers and the baseline for CFGD optimizer. Performance evaluation entails carrying out 20 runs for each pre-trained model with varying $\alpha$ values, and average accuracy is computed to create curves, as shown in Fig. 4. Evidently, the average accuracies gradually increased with rising fractional orders and peaked when $\alpha$ equaled 0.7. Thereafter, the curves exhibited a swift decline.

Table 2 displays the highest accuracy (Acc) achieved by each model when employing different optimization algorithms in this study to classify ocular tumors. It also encompasses their respective average computation times.

The CSA-CFGD algorithm consistently demonstrates the highest accuracy for all three pre-trained models. In particular, for Vgg16, it achieves an accuracy of 88.25% with a computation time of 2,973 seconds. AlexNet attains a high accuracy of 90.75% with a computation time of 798 seconds. On the other hand, with an accuracy of 91.75% and a computation time of 1,022 seconds, GoogLeNet outperforms other algorithms. Conversely, some alternative algorithms, such as ADAM and CSA, exhibit lower accuracies. These

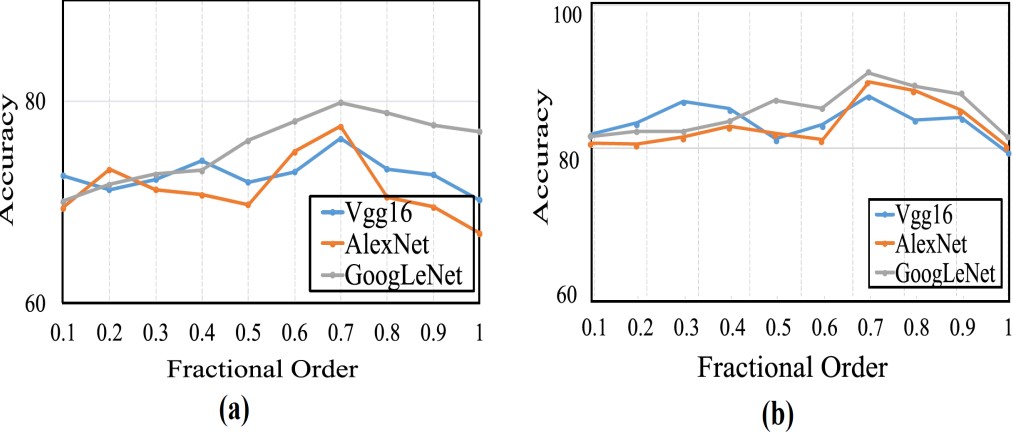

**Figure 4** The average accuracy for (A) CFGD and (B) CSA-CFGD with different values of α.

**Table 2** Performance metrics of CNN models with different optimizers.

| Optimizer | Vgg16 | | AlexNet | | GoogLeNet | |
|---|---|---|---|---|---|---|
| | Acc. | $T_{avg}$ | Acc. | $T_{avg}$ | Acc. | $T_{avg}$ |
| SGDM | 86.25 | 2,420 | 83.75 | 373 | 83.75 | 677 |
| ADAM | 78.5 | 2,442 | 72.5 | 444 | 82.5 | 728 |
| CSA | 77.26 | 2,382 | 79.55 | 324 | 82.81 | 618 |
| CFGD | 78.75 | 3,493 | 80.25 | 1,212 | 83.75 | 1,352 |
| BAS-ADAM | 80.5 | 3,098 | 89.25 | 935 | 85.5 | 1102 |
| CSA-ADAM | 87.25 | 3,062 | 87.5 | 940 | 88.75 | 1,094 |
| CSA-CFGD | 88.25 | 2,973 | 90.75 | 798 | 91.75 | 1,022 |

findings underscore the efficacy of the CSA-CFGD algorithm in attaining superior accuracy while maintaining reasonable computational efficiency for ocular tumor classification tasks using pre-trained models.

In Fig. 5, the comparison of optimization algorithms shows that the CSA-CFGD algorithm stands out with a higher level of stability and consistent superior performance with regard to accuracy. It impressively converges to better solutions, which then leads to the highest accuracy among all the tested algorithms. Meanwhile, the performance of ADAM exhibited constant improvement during the experiment, but it could not attain the level of accuracy achieved by CSA-CFGD. This, in turn, suggests that the CSA-CFGD algorithm possesses exceptional optimization capabilities, which enables it to deliver better results and outshine other algorithms in this specific ocular tumors classification task.

We performed 20 repetitions for each algorithm and evaluated various performance metrics to ensure the accuracy and comparability of our experiment. These metrics, such as the mean, best, worst, and standard deviation (Std) were recorded and summarized in Table 3.

The mean values provide insights into the average performance achieved by each algorithm over the repetitions. The best and worst values represent the highest and lowest

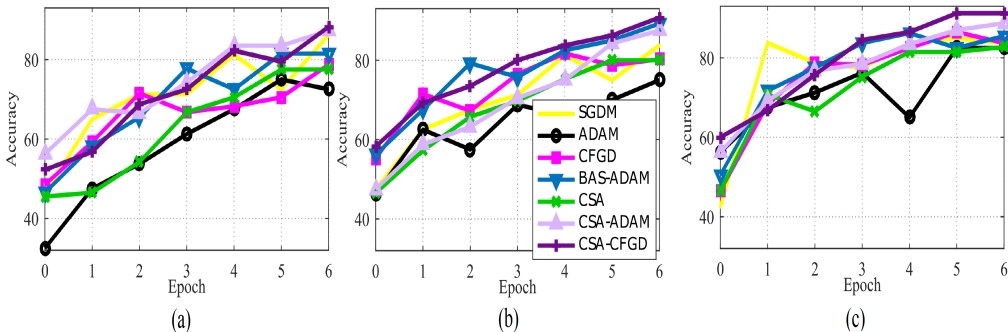

**Figure 5** The average accuracy of each optimizer for (A) Vgg16, (B) AlexNet, and (C) GoogLeNet.

**Table 3** The average performance metrics of CNN models using different optimizers.

| Optimizer | | Vgg16 | Alex | Google |
|---|---|---|---|---|
| SGDM | best | 86.25 | 83.75 | 83.75 |
| | worst | 52.50 | 72.50 | 70.00 |
| | mean | 74.38 | 77.07 | 78.00 |
| | Std | 7.31 | 3.59 | 3.70 |
| ADAM | best | 78.50 | 72.50 | 82.50 |
| | worst | 38.75 | 51.25 | 60.00 |
| | mean | 70.22 | 66.94 | 76.98 |
| | Std | 10.77 | 5.00 | 5.41 |
| CSA | best | 77.26 | 79.55 | 82.81 |
| | worst | 68.75 | 70.20 | 78.26 |
| | mean | 73.89 | 76.86 | 80.34 |
| | Std | 2.81 | 2.13 | 2.76 |
| CFGD | best | 78.75 | 80.25 | 83.75 |
| | worst | 73.50 | 72.00 | 75.00 |
| | mean | 76.29 | 77.51 | 79.83 |
| | Std | 1.58 | 2.26 | 2.64 |
| BAS-ADAM | best | 80.50 | 89.25 | 85.50 |
| | worst | 74.50 | 78.75 | 80.00 |
| | mean | 78.15 | 84.31 | 82.81 |
| | Std | 2.02 | 3.66 | 1.56 |
| CSA-ADAM | best | 87.25 | 87.50 | 88.75 |
| | worst | 78.75 | 79.00 | 80.25 |
| | mean | 82.79 | 82.96 | 85.36 |
| | Std | 2.59 | 2.81 | 2.56 |
| CSA-CFGD | best | 88.25 | 90.75 | 91.75 |
| | worst | 84.50 | 84.00 | 83.75 |
| | mean | 86.43 | 87.47 | 87.62 |
| | Std | 1.23 | 1.86 | 2.27 |

performance observed among the 20 runs, respectively. Moreover, Std provides useful information about the extent of variability in the results obtained, giving us a measure of the algorithm's consistency.

The obtained results in Table 3 provide insights into the performance of different optimization algorithms on Vgg16, AlexNet, and GoogLeNet models. For the SGDM algorithm, the average accuracy was 74.38% for Vgg16, 77.07% for AlexNet, and 78% for GoogLeNet, accompanied by relatively high standard deviation values of 7.31, 3.59, and 3.70, respectively. In contrast, the ADAM algorithm yielded lower average accuracy values of 70.22% for Vgg16, 66.94% for AlexNet, and 76.98% for GoogLeNet, with standard deviation values of 10.77, 5.00, and 5.41, respectively. Similarly, the CSA algorithm resulted in lower average accuracy values of 73.89% for Vgg16, 76.86% for AlexNet, and 80.34% for GoogLeNet, accompanied by standard deviation values of 2.81, 2.31, and 2.76, respectively.

The CFGD and BAS-ADAM algorithms showed higher mean accuracy of 76.29% and 78.15% for Vgg16, 77.51% and 84.31% for AlexNet, and 79.83% and 82.81% for GoogLeNet, respectively, with Std values ranging from 1.58 to 2.64. The CSA-ADAM algorithm exhibited better mean accuracy of 82.79% for Vgg16, 82.96% for AlexNet, and 85.36% for GoogLeNet, with relatively Std values of 2.59, 2.81, and 2.56, respectively. The CSA-CFGD algorithm demonstrated the highest mean accuracy values of 86.43% for Vgg16, 87.47% for AlexNet, and 87.62% for GoogLeNet, with relatively lower Std values of 1.23, 1.86, and 2.27. In summary, the CSA-CFGD algorithm demonstrated the most consistent and superior performance, featuring higher average accuracy and lower standard deviation across all three models.

Furthermore, the results show CSA-CFGD achieving the highest accuracy and ADAM showing the worst accuracy for Vgg16, AlexNet, and GoogLeNet. In particular, Vgg16 reaches its peak accuracy of 88.25% with CSA-CFGD and the worst accuracy of 38.75% with ADAM. In a similar vein, AlexNet attains its highest accuracy of 90.75% with CSA-CFGD and the worst accuracy of 51.25% with ADAM. GoogLeNet also demonstrates the highest accuracy of 91.75% with CSA-CFGD, whereas ADAM displays the worst accuracy of 60%.

Considering the pace at which each algorithm and model reaches convergence assumes significance. As shown in Fig. 6, CSA-CFGD stands out not only for reaching the minimum value, all the while doing so at a faster pace in comparison to the other optimization algorithms. While CSA-ADAM and BAS-ADAM come close to the lowest value, they fall just short of reaching it. CSA shows better performance when compared with SGDM and ADAM; however, it falls slightly short of reaching the lowest value as well. Conversely, SGDM, ADAM and CFGD are noticeably far from the lowest value. CSA-CFGD really shines among these approaches, achieving the lowest value with fewer iterations. In contrast, CSA-ADAM and BAS-ADAM require more iterations to get there.

Furthermore, the comparison of algorithmic performance often involves statistical hypothesis tests, such as the Wilcoxon rank-sum test. In our study, we applied this test to pairwise combinations of SGDM, ADAM, CSA, CFGD, BAS-ADAM, CSA-ADAM, and CSA-CFGD. The null hypothesis ($H_0$) assumes no significant difference in accuracy between the benchmark algorithm and CSA-CFGD for the proposed model. Rejection of

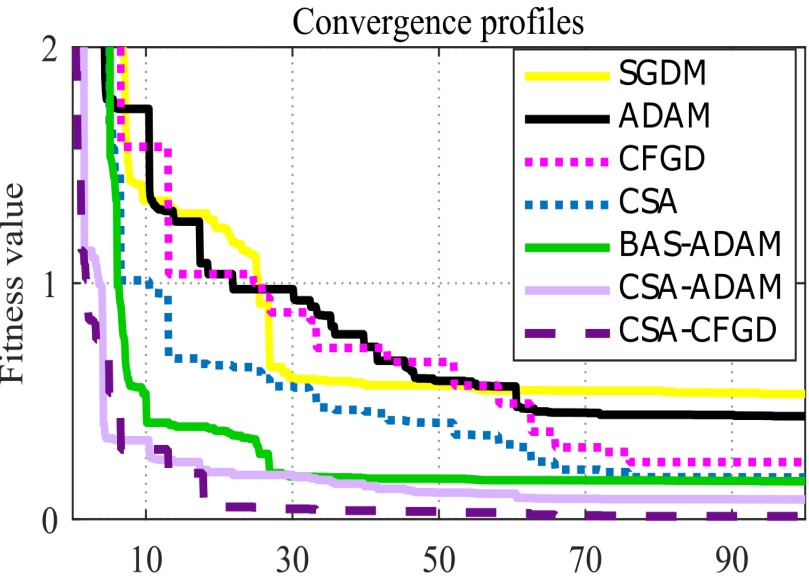

**Figure 6** Convergence profiles for an arbitrary epoch.

($H_0$) at a significance level ($a$) indicates a substantial difference in accuracy between the two populations, with a confidence level ($CL = (1 - a) \times 100\%$). The populations' sizes are denoted by $n_1$ and $n_2$, and their sum of ranks by $W_1$ and $W_2$, respectively. By comparing $W_1$ and $W_2$ to critical values from Wilcoxon rank-sum tables (*Montgomery & Runger, 2010*), we determined the rejection or acceptance of the null hypothesis. The results in Table 4 consistently reject the null hypothesis, affirming CSA-CFGD's superiority in ocular tumor classification.

Beside, the calculated $t$-value is compared to critical t-values for the chosen confidence level ($CL$). The null hypothesis assumes no significant group difference and is rejected if the $t$-value falls within the critical region, indicating significance. Conversely, acceptance of the null hypothesis occurs if the $t$-value falls outside the critical region. The $t$-test calculates the $t$-value using the formula:

$$t = \frac{W_1 - \mu}{SE}, \tag{19}$$

where $W_1$ is the sum of ranks for the first group, $\mu$ is the expected value of the sum of ranks under the null hypothesis ($n_1 * (n_1 + n_2 + 1)/2$), and $SE$ is the standard error of the ranks, given by:

$$SE = \sqrt{\frac{n_1 n_2 (n_1 + n_2 + 1)}{12}}. \tag{20}$$

To determine critical $t$-values for a given $CL$, we refer to the t-distribution table or use statistical software. Concerning a two-tailed test, we divide the chosen significance level ($a$) by 2. For example, if $CL = 0.05$, $a = 0.025$ for each tail. We find the critical $t$-value corresponding to $a = 0.025$ and degrees of freedom ($df = n_1 + n_2 - 2$). In a similar vein, for $CL = 0.01$, $a = 0.005$ for each tail; we find the corresponding critical $t$-value. By comparing

**Table 4 Statistical analysis of CNN models with respect to each optimizer.**

| Model | Optimizer | Two-sample $t$-test | | | Wilcoxon rank-sum test | | | | | | | |
|---|---|---|---|---|---|---|---|---|---|---|---|---|
| | | $t$-value | Accept/reject null hypothesis | CL | $n_1$ | $n_2$ | Critical value | | $W_1$ | $W_2$ | Accept/reject null hypothesis | CL |
| | | | | | | | 0.05 | 0.01 | | | | |
| Vgg16 | SGDM | 5.37E−07 | reject | 99% | 7 | 10 | 42 | 37 | 28 | 125 | reject | 99% |
| | | | | | 6 | 8 | 29 | 25 | 21 | 84 | reject | 99% |
| | | | | | 5 | 9 | 22 | 18 | 15 | 90 | reject | 99% |
| | ADAM | 1.63E−11 | reject | 99% | 7 | 10 | 42 | 37 | 28 | 125 | reject | 99% |
| | | | | | 6 | 8 | 29 | 25 | 21 | 84 | reject | 99% |
| | | | | | 5 | 9 | 22 | 18 | 15 | 90 | reject | 99% |
| | CSA | 1.30E−12 | reject | 99% | 7 | 10 | 42 | 37 | 28 | 125 | reject | 99% |
| | | | | | 6 | 8 | 29 | 25 | 21 | 84 | reject | 99% |
| | | | | | 5 | 9 | 22 | 18 | 15 | 90 | reject | 99% |
| | CFGD | 2.75E−15 | reject | 99% | 7 | 10 | 42 | 37 | 28 | 125 | reject | 99% |
| | | | | | 6 | 8 | 29 | 25 | 21 | 84 | reject | 99% |
| | | | | | 5 | 9 | 22 | 18 | 15 | 90 | reject | 99% |
| | BAS-ADAM | 9.52E−13 | reject | 99% | 7 | 10 | 42 | 37 | 28 | 125 | reject | 99% |
| | | | | | 6 | 8 | 29 | 25 | 21 | 84 | reject | 99% |
| | | | | | 5 | 9 | 22 | 18 | 15 | 90 | reject | 99% |
| | CSA-ADAM | 2.25E−05 | reject | 99% | 7 | 10 | 42 | 37 | 28 | 125 | reject | 99% |
| | | | | | 6 | 8 | 29 | 25 | 21 | 84 | reject | 99% |
| | | | | | 5 | 9 | 22 | 18 | 15 | 90 | reject | 99% |
| AlexNet | SGDM | 1.36E−08 | reject | 99% | 7 | 10 | 42 | 37 | 28 | 125 | reject | 99% |
| | | | | | 6 | 8 | 29 | 25 | 21 | 84 | reject | 99% |
| | | | | | 5 | 9 | 22 | 18 | 15 | 90 | reject | 99% |
| | ADAM | 9.47E−13 | reject | 99% | 7 | 10 | 42 | 37 | 28 | 125 | reject | 99% |
| | | | | | 6 | 8 | 29 | 25 | 21 | 84 | reject | 99% |
| | | | | | 5 | 9 | 22 | 18 | 15 | 90 | reject | 99% |
| | CSA | 4.28E−09 | reject | 99% | 7 | 10 | 42 | 37 | 28 | 125 | reject | 99% |
| | | | | | 6 | 8 | 29 | 25 | 21 | 84 | reject | 99% |
| | | | | | 5 | 9 | 22 | 18 | 15 | 90 | reject | 99% |
| | CFGD | 3.41E−13 | reject | 99% | 7 | 10 | 42 | 37 | 28 | 125 | reject | 99% |
| | | | | | 6 | 8 | 29 | 25 | 21 | 84 | reject | 99% |
| | | | | | 5 | 9 | 22 | 18 | 15 | 90 | reject | 99% |
| | BAS-ADAM | 0.0055 | reject | 99% | 7 | 10 | 42 | 37 | 43 | 110 | accept | |
| | | | | | 6 | 8 | 29 | 25 | 32 | 73 | accept | |
| | | | | | 5 | 9 | 22 | 18 | 28 | 77 | accept | |
| | CSA-ADAM | 6.98E−06 | reject | 99% | 7 | 10 | 42 | 37 | 28 | 125 | reject | 99% |
| | | | | | 6 | 8 | 29 | 25 | 21 | 84 | reject | 99% |
| | | | | | 5 | 9 | 22 | 18 | 15 | 90 | reject | 99% |

**Table 4** (*continued*)

| Model | Optimizer | Two-sample *t*-test | | | Wilcoxon rank-sum test | | | | | | | | |
|---|---|---|---|---|---|---|---|---|---|---|---|---|---|
| | | *t*-value | Accept/reject null hypothesis | CL | $n_1$ | $n_2$ | Critical value | | $W_1$ | $W_2$ | Accept/reject null hypothesis | CL |
| | | | | | | | 0.05 | 0.01 | | | | |
| GoogLeNet | SGDM | 1.46E−10 | reject | 99% | 7 | 10 | 42 | 37 | 28 | 125 | reject | 99% |
| | | | | | 6 | 8 | 29 | 25 | 21 | 84 | reject | 99% |
| | | | | | 5 | 9 | 22 | 18 | 15 | 90 | reject | 99% |
| | ADAM | 6.30E−07 | reject | 99% | 7 | 10 | 42 | 37 | 28 | 125 | reject | 99% |
| | | | | | 6 | 8 | 29 | 25 | 21 | 84 | reject | 99% |
| | | | | | 5 | 9 | 22 | 18 | 15 | 90 | reject | 99% |
| | CSA | 3.31E−11 | reject | 99% | 7 | 10 | 42 | 37 | 28 | 125 | reject | 99% |
| | | | | | 6 | 8 | 29 | 25 | 21 | 84 | reject | 99% |
| | | | | | 5 | 9 | 22 | 18 | 15 | 90 | reject | 99% |
| | CFGD | 8.78E−09 | reject | 99% | 7 | 10 | 42 | 37 | 28 | 125 | reject | 99% |
| | | | | | 6 | 8 | 29 | 25 | 21 | 84 | reject | 99% |
| | | | | | 5 | 9 | 22 | 18 | 15 | 90 | reject | 99% |
| | BAS-ADAM | 1.20E−08 | reject | 99% | 7 | 10 | 42 | 37 | 29 | 124 | reject | 99% |
| | | | | | 6 | 8 | 29 | 25 | 22 | 83 | reject | 99% |
| | | | | | 5 | 9 | 22 | 18 | 16 | 89 | reject | 99% |
| | CSA-ADAM | 9.85E−05 | reject | 99% | 7 | 10 | 42 | 37 | 43 | 110 | accept | |
| | | | | | 6 | 8 | 29 | 25 | 35 | 70 | accept | |
| | | | | | 5 | 9 | 22 | 18 | 29 | 76 | accept | |

the calculated *t*-value with critical *t*-values at the chosen *CL*, we determine whether the null hypothesis can be accepted or rejected.

The outcomes presented in Table 4 indicate that CSA-CFGD is the most accurate optimization tool for the proposed model, especially concerning magnitude responses. This underscores its effectiveness in optimizing the model's performance.

## DISCUSSION

In this section, we emphasize the key insights drawn from our findings. Our experiments facilitated the examination of various evaluation metrics, revealing that the CSA-CFGD algorithm consistently outperformed competing methods in terms of both mean accuracy and stability across each model. Particularly noteworthy is the exceptional performance of the CSA-CFGD algorithm when paired with the GoogLeNet model, achieving an impressive mean accuracy of 87.62%. In contrast, rival algorithms in this model exhibited mean accuracies ranging from 66.94% to 87.47%. Despite the higher standard deviation (Std) of GoogLeNet with CSA-CFGD compared to GoogLeNet with BAS-ADAM, its accuracy remains superior to that of BAS-ADAM. This suggests that GoogLeNet with CSA-CFGD may exhibit variability across different runs compared to BAS-ADAM, highlighting its unique performance characteristics.

For each pre-trained model, Fig. 6 depicts an outstanding convergence between the results of CSA-CFGD and BAS-ADAM, which, in turn, indicates that both optimization

algorithms reach similar solutions and corresponding levels of accuracy during the training. Both CSA-CFGD and BAS-ADAM are shown to be efficient at improving the trained models and achieving high classification accuracies for ocular tumor images. However, CSA-CFGD is more steady and stable in reaching the optimal solution. The convergence shown in Fig. 6 deepens our comprehension of the optimization process as a whole and offers vital insights into how well-suited these algorithms are to the specific task at hand.

The statistical evidence provides strong confidence in the reliability of the findings, thus reinforcing the argument to adopt CSA-CFGD as the preferred optimization algorithm for such tasks. The hypothesis test results shown in Table 4 consistently reject the null hypothesis in the majority of cases. CSA-CFGD outperforms SGDM, ADAM, CSA, CFGD, BAS-ADAM and CSA-ADAM in ocular tumor classification, indicating its superiority.

The suggested approach investigates the incorporation of fractional calculus principles into the CSA to enhance the optimization strategy using CFGD. This research suggests that while this method achieves rapid convergence, it demands extended training durations. However, leveraging historical information through memory retention could enhance robustness, enabling the algorithm to navigate noisy or challenging optimization scenarios more effectively. The integration offers adaptability and robustness, yet these advantages pose challenges in terms of implementation, scalability, and generalizability across diverse optimization scenarios, requiring further exploration. Furthermore, a reliance on CFGD in later stages occasionally yields non-optimal solutions, impacting exploration strategies and the method's consistency in reaching optimal solutions.

Our proposed method has limitations that span various facets, such as computational overhead stemming from the intricate nature of the approach, algorithmic complexity resulting from the integration of multiple techniques, and the potential for domain-specific efficacy, which may limit its applicability to specific problem domains. It is imperative to address these limitations to enhance the method's performance and broaden its applicability across diverse optimization landscapes.

## CONCLUSION

In this article, we proposed a hybrid approach integrating Caputo fractional gradient descent with cuckoo search so as to create an efficient optimization algorithm. This hybrid method is applied to train pre-trained models (Vgg16, AlexNet, and GoogLeNet) for ocular tumor diagnosis. Our proposed optimizer is compared with various existing optimizers in the literature, including SGDM, ADAM, CSA, CFGD, BAS-ADAM and CSA-ADAM. The outcomes illustrate the substantial potential of our optimizer in enhancing classification accuracy and accelerating convergence speed. Notably, AlexNet achieved the swiftest training time at 798 seconds compared to unconventional optimization approaches (CFGD, BAS-ADAM, and CSA-ADAM), while GoogLeNet demonstrated the highest accuracy of 87.62% across 20 multiple rounds. Our proposed algorithm consistently attained the highest accuracy across all three models and exhibited lower standard deviations, indicating stable performance. Remarkably, the mean accuracy reached by the proposed optimizer was 86.43%, 87.42%, and 87.62% for the Vgg16, AlexNet, and GoogLeNet models, respectively.

Furthermore, statistical hypothesis tests consistently affirmed the performance superiority of our algorithm over other alternatives. These results underscore the efficacy of our optimization algorithm in enhancing the performance of pre-trained models for ocular tumor diagnosis.

In future research endeavors, this work could be further enhanced by incorporating feature selection methods to identify pertinent features, employing image enhancement techniques to enhance image quality, and exploring hybrid algorithms that amalgamate the strengths of various optimization approaches. Additionally, expanding the dataset to encompass a broader range of ocular tumor cases could facilitate a more comprehensive evaluation of performance, contributing to more accurate and efficient diagnoses in real-world clinical settings. Furthermore, the consideration of training the models from scratch using the proposed optimizer should be explored, given its potential impact on achieving superior performance. This would involve carrying out comprehensive experiments to determine the optimal configuration of the $\alpha$-order parameter by considering factors such as the efficiency and rapid convergence observed during the training process. Furthermore, investigating the behavior of the algorithm in comparison to existing optimization techniques can yield valuable insights into its effectiveness across various applications and datasets.

## ACKNOWLEDGEMENTS

We would like to express our appreciation to the Retina Image Bank for providing access to their dataset, which significantly contributed to our research.

### Funding

This project was supported by National Natural Science Foundation of China No.62172152. The funders had no role in study design, data collection and analysis, decision to publish, or preparation of the manuscript.

### Grant Disclosures

The following grant information was disclosed by the authors:
National Natural Science Foundation of China: 62172152.

### Competing Interests

The authors declare there are no competing interests.

### Author Contributions

- Abduljlil Abduljlil Ali Abduljlil Habeb conceived and designed the experiments, performed the experiments, analyzed the data, performed the computation work, authored or reviewed drafts of the article, and approved the final draft.
- Ningbo Zhu analyzed the data, authored or reviewed drafts of the article, and approved the final draft.

- Mundher Mohammed Taresh performed the computation work, prepared figures and/or tables, and approved the final draft.
- Talal Ahmed Ali Ali conceived and designed the experiments, authored or reviewed drafts of the article, revised the mathematical formula used in the study, and approved the final draft.

### Data Availability

The ocular tumors data was collected from the Retina Image Bank®:
https://imagebank.asrs.org.

### Supplemental Information

Supplemental information for this article can be found online at http://dx.doi.org/10.7717/peerj-cs.1923#supplemental-information.

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
