# Peer review of "Deep ocular tumor classification model using cuckoo search algorithm and Caputo fractional gradient descent"

_PeerJ Computer Science, doi:10.7717/peerj-cs.1923_

## Round 0.1 · original submission · Major Revisions

· Academic Editor

Major Revisions

Dear authors,

Thank you for submitting your article. Reviewers have now commented on your article and suggested major revisions. When submitting the revised version of your article, it will be better to address the followings:

1- The research gaps and contributions should be clearly summarized in the introduction section. Please evaluate how your study is different from others in the related section.

2- The values for the parameters of the algorithms selected for comparison should be given.

3- Clarifying the study’s limitations allows the readers to better understand under which conditions the results should be interpreted. A clear description of the limitations of a study also shows that the researcher has a holistic understanding of his/her study. However, the authors fail to demonstrate this in their paper. The authors should clarify the pros and cons of the methods. What are the limitation(s) methodology(ies) adopted in this work? Please indicate practical advantages, and discuss research limitations.

4- Explanation of the equations should be checked. Equations should be used with correct equation numbers within the text.

Best wishes,

**Language Note:** The review process has identified that the English language must be improved. PeerJ can provide language editing services - please contact us at [email protected] for pricing (be sure to provide your manuscript number and title). Alternatively, you should make your own arrangements to improve the language quality and provide details in your response letter. – PeerJ Staff

·

Basic reporting

The paper is well-structured, written in a standardized manner, clearly described, and the data can be obtained from the official website, which meets the requirements of the journal. By summarizing the previous work, he put forward his own point of view, that is, to improve the cuckoo search algorithm by using fractional differentiation to improve the computational accuracy and convergence speed.

Experimental design

The experimental design is reasonable, and the performance of the new optimizer is compared with the existing method, and the results are good.

Validity of the findings

The data can be obtained from the official website.

·

Basic reporting

Although the authors have done a nice job for the chosen research problem but still the incorporation of the following points will still help in enhancing its quality:

1) Revise the abstract and make it perfect so that the reader can easily understand your work.
2) What is the significance of the fractional order gradient descent algorithm? Also, explain the complexity of the algorithm will this increase by combining the CSA and CFGD? In addition, the author should explain about the convergence of the algorithm. At which fractional order the algorithm shows fast convergence.
3) Improve the grammar of the whole paper.
4) Which activation function you have used in your work and what was the reason for choosing that?
5) Revise Table 1. Add all the parameters used in your study.
6) Is your proposed algorithm a modified one or a hybrid one?

Experimental design

no comment

Validity of the findings

no comment

Reviewer 3 ·

Basic reporting

In this manuscript, the authors proposed a new variant of the algorithm cuckoo search algorithm using the Caputo fractional gradient descent and applied to solve an ocular tumor classification problem. The experiments on a dataset of 400 fundus images on pre-trained models showed the advantages of algorithms over the existing algorithms SGDM, ADAM, CFGD, BAS-ADAM and CSA-ADAM. The proposed algorithm that combines the fractional descent method with meta-heuristic algorithms is novel and interesting. However, the following issues should be addressed before publication:
1. The manuscript should be carefully checked as there are some formula labeling problems. For example, label (1) should be (4) in Line 128, label (2) should be (4) in Line 132,label (8) should be (10) in Algorithm 2.
2. Formula (6) is the key formula and should be derived in detail. Each parameter should be explained.
3. The authors should attempt to explain why the fractional gradient can improve the performance of the cuckoo search algorithm.
4. In the experiments, the hyper-parameters in table 1 are not clear enough. What is the relationship between the value of Epochs and that of Iteration? What kind of loss function was used during the training process. The authors should give the specific value of the fractional parameters α of the Caputo fractional gradient, and indicate how to select the fractional parameters α and how the fractional order parameters affect experimental results.
5. The algorithm comparisons are all based on pre-trained models. Does this method have any advantages when training from scratch?
6. Formula (6) shows the new algorithm needs more gradient computations. Why is there an advantage in computation speed?

Experimental design

In the experiments, the hyper-parameters in table 1 are not clear enough. What is the relationship between the value of Epochs and that of Iteration? What kind of loss function was used during the training process. The authors should give the specific value of the fractional parameters α of the Caputo fractional gradient, and indicate how to select the fractional parameters α and how the fractional order parameters affect experimental results.
The algorithm comparisons are all based on pre-trained models. Does this method have any advantages when training from scratch?

Validity of the findings

no comment

---

## Round 0.2 · accepted · Accept

· Academic Editor

Accept

Dear authors,

Thank you for the revision. The paper seems to be improved in the opinion of the reviewers. The paper is now ready to be published.

Best wishes,

·

Basic reporting

no comment

Experimental design

no comment

Validity of the findings

no comment

Additional comments

The author did a good job revising my comments and I have no issues anymore.

·

Basic reporting

The authors incorporate all the points.

Experimental design

good

Validity of the findings

yes